# Evaluation of Retinal Changes in Women with Different Phenotypes of Polycystic Ovary Syndrome

**DOI:** 10.3390/diagnostics15020227

**Published:** 2025-01-20

**Authors:** Fatma Sumer, Beril Gurlek, Elif Yildiz, Feyzahan Uzun, Sabri Colak, Ilknur Merve Ayazoglu

**Affiliations:** 1Department of Ophthalmology, Faculty of Medicine, Recep Tayyip Erdogan University, Rize 53100, Turkey; feyzahan@gmail.com; 2Department of Obsetrics and Gynaecology, Faculty of Medicine, Recep Tayyip Erdogan University, Rize 53100, Turkey; beril.gurlek@erdogan.edu.tr (B.G.); dr.sabricolak@gmail.com (S.C.); ilknurmervekazaz@gmail.com (I.M.A.); 3Department of Obsetrics and Gynaecology, İstanbul Gaziosmanpasa Training and Research Hospital, Istanbul 34255, Turkey; elifsumer_@hotmail.com

**Keywords:** choroidal thickness, macular thickness, optical coherence tomography, polycystic ovary syndrome, retinal nerve fiber layer

## Abstract

**Background**: The aim of our study was to evaluate the retinal nerve fiber layer (RNFL) and macular and choroidal thicknesses in women with different phenotypes of polycystic ovary syndrome (PCOS), and compare these measurements with those of healthy women of reproductive age. **Materials and Methods**: This prospective case–control study included 120 eyes of 120 women with PCOS, with each of the four distinct phenotypes comprising 30 eyes of 30 women. Additionally, 30 eyes from 30 healthy women were included in the control group. All participants underwent comprehensive ophthalmologic examinations, and RNFL thickness, macular thickness (MT), and choroidal thickness (CT) in each eye were measured via spectral-domain optical coherence tomography. The body mass index (BMI) of the patients was recorded and compared with the retinal changes. **Results**: The average mean and nasal segments of the RNFL were significantly greater in the PCOS group than in the healthy control group (*p* < 0.001). There was a statistically significant difference in foveal retinal thickness between the groups (*p* < 0.001). Our study revealed significant choroidal tissue thickening subfoveally and at locations 500 μm temporal, 500 μm nasal, 1500 μm nasal, and 1500 μm temporal to the fovea in all phenotypes of the PCOS group (*p* < 0.001). Additionally, there was a positive correlation between BMI and all CT changes. **Conclusions**: Our findings indicate that the retinal layers and choroid are affected by all phenotypes of PCOS, one of the most common reproductive abnormalities, albeit to varying degrees. Furthermore, these changes were found to be correlated with BMI.

## 1. Introduction

Polycystic ovary syndrome (PCOS) is considered an endocrine disorder that affects women during their reproductive years and is one of the causes of difficulty in becoming pregnant in the reproductive age [1]. Its clinical presentation is quite heterogeneous and is characterized by menstrual irregularities or amenorrhea, increased androgen levels and a PCOS-specific appearance that can be visualized by ultrasonography in the ovaries [2]. In addition to its reproductive implications, PCOS is associated with an increased risk of obesity, insulin resistance, metabolic syndrome, and cardiovascular diseases. The interplay between insulin resistance and hyperinsulinemia is particularly noteworthy, as these factors lead to heightened androgen production and diminished levels of sex hormone-binding globulin, thereby exacerbating hyperandrogenism [3]. The 2012 National Institutes of Health (NIH) Evidence-Based Methodology Workshop on PCOS proposed a refined classification system that delineates PCOS into four distinct phenotypes, building upon the Rotterdam criteria established in 2003. This classification framework is intended to enhance the understanding of PCOS, facilitate the identification of the associated risk factors, improve clinical management strategies, and clarify the systemic effects of the disorder [4].

Recent investigations have revealed the presence of sex hormone receptors in various ocular tissues, including the cornea, conjunctiva, iris, ciliary body, lacrimal and meibomian glands, as well as the optic nerve and retinochoroidal structures [5]. The retina possesses the capability to synthesize neurosteroids from cholesterol, while estrogens and androgens produced in the gonads and adrenal glands are delivered to the ocular tissues via the systemic circulation. Emerging evidence suggests that physiological levels of estrogen confer beneficial and protective effects on neuroretinal function [6]. Testosterone, acting on neural androgen receptors, has been shown to promote myelin regeneration and exert neuroprotective effects on the optic nerve [7].

Furthermore, it has been posited that estrogen enhances ocular blood flow through vasodilation, whereas testosterone may exert antagonistic effects on ocular blood flow relative to estrogen [8]. The influence of sex hormones on the development of various retinal and optic nerve disorders such as age-related macular degeneration, optic neuritis, glaucoma, central serous chorioretinopathy, and retinitis pigmentosa has been hypothesized [9]. Factors such as age, physiological hormonal fluctuations throughout the menstrual cycle, pregnancy, menopause, and pathological hormonal states, including those observed in PCOS, may impact the distribution and configuration of gonadal hormone receptors in ocular tissues, leading to variations in ocular physiopathology [5]. Consequently, these factors have been associated with a range of conflicting findings in the existing literature.

Numerous studies have investigated alterations in macular, choroidal, and retinal nerve fiber layer (RNFL) thickness in patients with PCOS compared to healthy controls of reproductive age. However, the significance of the different subphenotypes of PCOS in this context remains underexplored. The current study compares the alterations in RNFL, macula, and choroid between patients with different PCOS phenotypes and healthy persons using spectral-domain optical coherence tomography (SD-OCT). The connections between these ocular results and hormonal alterations will also be assessed.

## 2. Methods

Prior to the commencement of the study, formal approval was obtained from the institutional ethical committee for this prospective case–control study, which was conducted at a tertiary university hospital. The researchers diligently adhered to the guidelines set forth in the Declaration of Helsinki, and each participant graciously provided their written informed consent.

The study enrolled 120 women diagnosed with PCOS, differentiated by subphenotype, at the Obstetrics and Gynecology Clinic from August 2020 to July 2021. Additionally, 30 healthy women with regular menstrual cycles and normal ovarian morphology, employed at the same hospital, were recruited as control participants. All participants were confirmed to be non-pregnant. Comprehensive medical data and personal medical histories were collected from each participant, and the body mass index (BMI) was calculated using the formula weight (kg) divided by height (m^2^).

### 2.1. PCOS Diagnosis

The diagnosis of polycystic ovary syndrome (PCOS) was conducted in accordance with the Rotterdam 2003 criteria, which consist of three essential components: (1) oligomenorrhea, defined as menstrual cycles occurring at intervals greater than 35 days or the occurrence of eight or fewer menstrual cycles within a year; (2) clinical and/or biochemical indicators of hyperandrogenism, which may manifest through clinical signs such as hirsutism, acne, and androgenic alopecia, or through laboratory assessments revealing elevated serum levels of both total and free testosterone; and (3) ultrasonographic evidence of polycystic ovaries, characterized by the presence of 12 or more follicles measuring between 2 and 9 mm in diameter, and/or an ovarian volume exceeding 10 mL. The identification of a single ovary meeting these specified criteria was considered adequate for establishing a diagnosis of PCOS.

Subsequent to the diagnosis, participants were classified into one of four distinct phenotypes based on their clinical presentation and laboratory findings:-**Phenotype A:** This phenotype included all three diagnostic criteria: menstrual cycle irregularity, blood and clinical detection of hyperandrogenism, and ultrasound appearance of polycystic ovaries.-**Phenotype B:** Individuals in this category exhibited menstrual irregularity and hyperandrogenism, but lacked ultrasound evidence of polycystic ovaries.-**Phenotype C:** This phenotype was characterized by the presence of hyperandrogenism, alongside the ultrasound identification of polycystic ovaries.-**Phenotype D:** Participants classified under this phenotype displayed menstrual irregularity and ultrasound evidence of polycystic ovaries, yet did not present with hyperandrogenism.

Patients were systematically categorized into one of these four phenotypes based on the established diagnostic criteria, and their pertinent information was meticulously documented for further analysis.

### 2.2. Clinical Assessments

Participants diagnosed at the Gynecology and Obstetrics Clinic underwent ultrasound imaging. Venous blood samples required for hormone evaluation were taken between the 3rd and 5th days of menstruation, or between the 3rd and 5th days of withdrawal bleeding after oral progesterone treatment in patients whose menstrual cycles could not be determined due to menstrual irregularity. Venous samples were taken from the antecubital vein in a sitting position between 08:00 and 09:00 in the morning, and the participants were fasted for 12 h before sample collection. Luteinizing hormone (LH), follicle stimulating hormone (FSH), estradiol, and both total and free testosterone concentrations were then measured.

### 2.3. Ophthalmological Examination

After the preliminary evaluations, the subjects were sent to the ophthalmology clinic, where no particular phenotype was found. Clinical ocular examinations were performed by the same physician (blinded to PCOS phenotypes (FS)) at the Faculty of Ophthalmology, Recep Tayyip Erdogan University. A thorough ophthalmological examination was performed on each subject, encompassing fundus examination, biomicroscopy, tonometry, and best-corrected visual acuity assessments. After the dilation of the pupils with 0.5% tropicamide and 5% phenylephrine hydrochloride, fundoscopic examination, fundus photography, and macular and choroidal thickness measurements were performed using optical coherence tomography (SD-OCT, software version 5.6.3.0; Heidelberg, Germany) to determine retinal nerve fiber layer (RNFL) thickness. In the macular SD-OCT assessment, central and mean layer thicknesses were analyzed by scanning a 6 × 6 mm area using the Early Treatment Diabetic Retinopathy Study (ETDRS) macular scan, which divides the macula into nine ETDRS areas, including a central 1 mm circle and inner and outer rings with diameters of 3 mm and 6 mm, respectively. RNFL thickness was meticulously measured in six quadrants: superior temporal, temporal, inferior temporal, superior nasal, nasal, and inferior nasal, utilizing a circular scanning pattern centered on the optic nerve. Choroidal thickness measurements were recorded from the nasal and temporal margins at 500 μm and 1500 μm from the foveal center. Choroidal thickness was measured manually by measuring the distance between the outer edge of the hyperreflective band of the retinal pigment epithelium and the inner border of the scleral hyperreflectivity. This measurement was performed between 14:00 and 17:00 to avoid the previously reported diurnal variation in choroidal thickness in all participants. All measurements were performed by 2 different trained ophthalmic technicians blinded to the participants’ study group assignment.

### 2.4. Exclusion Criteria

Subjects who had utilized hormonal medications within the preceding six months or had congenital or acquired endocrine disorders were excluded from the study. Furthermore, individuals with any corneal, lens, or ocular surface abnormalities; a history of glaucoma, ocular surgery, or trauma; keratorefractive surgery; refractive errors exceeding 2 diopters; or a history of contact lens use were also excluded.

### 2.5. Statistical Analysis

Data analysis was performed utilizing the IBM SPSS 23.0 statistical software package. The assessment of normality in the distribution of data, both within and between groups, was conducted through the application of the Kolmogorov–Smirnov test and the Shapiro–Wilk test. For the comparison of non-normally distributed data across binary groups, the Mann–Whitney U test was utilized. In contrast, normally distributed data were subjected to a one-way analysis of variance (ANOVA). For non-normally distributed data across multiple groups, the Kruskal–Wallis H test was employed. Categorical data were analyzed using Fisher’s exact test, and adjustments for multiple comparisons were made using the Z test with Bonferroni correction. The relationships between quantitative variables that did not conform to a normal distribution were assessed using Spearman’s rank correlation coefficient (Spearman’s rho). The results of these analyses are reported as the mean ± standard deviation for normally distributed quantitative data, the median (minimum–maximum) for non-normally distributed quantitative data, and frequency (percentage) for categorical data. A significance threshold of *p* < 0.05 was established for all statistical tests conducted.

## 3. Results

The study was started with 165 patients, but during the study, 9 patients with adenoviral conjunctivitis, 2 patients with uveitis attack, and 4 patients in the POF group were excluded from the study because they withdrew from the study. This study was conducted with 150 participants, including 120 women diagnosed with PCOS divided into 4 separate phenotype groups and 30 healthy women as the control group. Only the right eyes of 150 female individuals were included. All patients in both PCOS and control groups had 1.0 Snellen visual acuity, and their anterior and posterior segment examinations were normal. Although intraocular pressures were higher in the PCOS group (20 (19–24)) compared to the control group (18 (16–21)), it was not statistically significant (*p* = 0.321). The mean age was 27.2 ± 6.28 years in the PCOS group and 29.08 ± 4.96 years in the control group, with no significant difference in terms of age between the groups (*p* = 0.07). The BMI and serum progesterone (PG), estradiol, and free testosterone levels were compared between the groups. The serum estradiol level, free testosterone level, and BMI were significantly greater in the PCOS group than in the control group (*p* < 0.001), whereas PG was not significantly different between the groups. The demographic characteristics of the groups are shown in Table 1.

The peripapillary RNFL, macular and choroidal thickness analyses and macular volume measurements are presented in Table 2, Table 3 and Table 4. The average mean (global: G) and nasal quadrant RNFL were significantly greater in the PCOS group than in the healthy control group (*p* < 0.001). When all retinal areas, including the macula, were compared between the PCOS patients and the control group, the nasal outer macula (NOM) was significantly greater in all PCOS phenotypes than in the control group (*p* < 0.001). In contrast to those in the nasal region, the temporal inner macula (TIM) and temporal outer macula (TOM) in all PCOS phenotypes were significantly thinner than those in the control group (*p* < 0.001). Additionally, the central part of the macula in the PCOS group was also found to be significantly thinner than that in the control group (*p* = 0.06).

CT assessments were performed at five locations, and all PCOS phenotypes were compared with those of the control group. The CT in all PCOS phenotypes was significantly greater than that in the control group at all measured locations (*p* < 0.001). When the mean macular volume was compared, the central subfield (CSF), temporal inner macula (TIM), and temporal outer macula (TOM) were significantly thinner in the PCOS group than in the control group (*p* < 0.001). Conversely, the nasal outer macula (NOM) was significantly thicker in the PCOS group than in the control group (*p* < 0.001).

The mean CT values for SCT, CT nasal 500 μm, CT temporal 500 μm, CT nasal 1500 μm, and CT temporal 1500 μm at five locations, along with the correlation of foveal thickness with BMI, estradiol, and free testosterone levels, are summarized in Table 5. Pearson correlation analysis revealed a negative correlation between BMI and CCT, and subfoveal thickness.

## 4. Discussion

PCOS, also known as ovarian hyperandrogenism, is a widely prevalent endocrine disorder among women of reproductive age. The utilization of various diagnostic criteria has resulted in variations in reported PCOS prevalence. The first study using the Rotterdam diagnostic criteria revealed a prevalence of 17%, leading to the classification of PCOS into different phenotypes [10]. PCOS is not only a disease that causes menstrual irregularity, but is linked to many metabolic conditions such as obesity, hypertension, dyslipidemia, insulin resistance, atherosclerosis, cardiovascular disease, impaired glucose tolerance, and diabetes. Elevated sex steroid levels in individuals with PCOS influence ocular structures, blood flow, and physiology in a manner comparable to their effects on the cardiovascular system and endometrium [11].

Although the impact of sex steroids on the optic nerve is not clear, previous studies have suggested that estrogen and testosterone may have trophic and neuroprotective effects and contribute to the regulation of ocular blood flow [7]. Estrogens slow neuronal injury and apoptotic pathways, and support the expression of cell survival genes, whereas testosterone acts via androgen receptors in the nervous system and has antioxidant and antiapoptotic potential [12]. Souza-Junior et al. reported notable thickening of the superior RNFL around the optic nerve in individuals with PCOS compared with healthy controls [13]. Similarly, Shriomani et al. detected an increased RNFL thickness in the superior quadrant among PCOS patients, especially in those with a body mass index greater than 30 kg/m^2^ [14]. Furthermore, Alpogan et al. reported that the average RNFL thickness in the PCOS group exceeded that in the healthy female cohort across all quadrants, although the differences were not statistically significant [15]. Consistent with these findings, our study revealed that RNFL thickness was significantly greater in women with all phenotypes of PCOS than in healthy women, especially in the superior nasal, inferior nasal, and nasal quadrants.

In their comprehensive assessment, Açmaz et al. utilized SD-OCT to evaluate the macula thickness, RNFL thickness, and choroidal thickness in PCOS patients. Their findings indicated that the temporal outer macula (TOM) and nasal outer macula (NOM) were significantly thicker in the PCOS group than in the control group. Conversely, the central foveal thickness and temporal inner macula were notably thinner in the PCOS group. Researchers have suggested that the observed increase in choroidal thickness and RNFL may contribute to the increased retinal thickness and volume in the peripheral retina [16]. However, they did not examine the alterations through different PCOS phenotypes on the retina and choroid, as classified according to the 2003 Rotterdam criteria. In our study, we demonstrated that the central volume and thickness of the TIM, TOM, and foveal areas decreased in patients with all the phenotypes of PCOS. Although this decrease was significant in all PCOS phenotypes compared with the control group, it was more pronounced in phenotype A. However, the peripheral part of the retina, such as the NOM, was significantly greater in patients with all PCOS phenotypes. Additionally, the nasal (N) and global (G) parts of the RNFL, and the choroid layer were significantly thicker in the PCOS group than in the control group. One way to interpret these findings is to suggest that increased choroidal and RNFL thickness may contribute to increased retinal volume and thickness on the nasal side of the retina. Although these data are similar to the results of Açmaz et al., we found that the classical PCOS findings are most consistent with phenotype A [16]. Although there have been studies evaluating neurodegeneration and optic nerve vascularity with OCTA, phenotyping has not been performed. These studies have shown that neurodegeneration or subclinical retinal inflammation due to insulin resistance tends to result in thinner GCL, IPL, and PRL compared to healthy, age-matched controls [17,18].

Sex hormones may also affect tissue perfusion by regulating blood flow in the retina and choroid. While estrogens exert a vasodilatory effect on retinal perfusion by reducing vascular resistance, testosterone, like progesterone, has the opposite effect [6]. Ornek et al. reported that ocular blood flow velocity is increased in PCOS patients and that vascular resistance seems to decrease only in the ophthalmic artery [11]. Choroidal thickness (CT) is influenced by a variety of factors, including age, sex, hormonal status, systemic and local diseases, as well as diurnal variations. The choroid plays a crucial role in supplying the majority of the retina’s nutritional needs and is characterized by the highest blood flow per unit weight in the human body. Consequently, alterations in choroidal thickness, which are affected by blood flow dynamics and ocular perfusion pressure, may serve as indicators of various physiological and pathological ocular conditions [19]. It has been reported that subfoveal CT in PCOS patients is significantly greater than that in control patients. CT was moderately correlated with free testosterone and estradiol levels. Researchers have concluded that the increase in CT is due to the unopposed vasodilatory effect of elevated estrogen [20]. In our research, although phenotype A was the most prevalent PCOS phenotype, we observed that the CT was greater in all quadrants than in the control group, corroborating previous findings.

PCOS is widely recognized for its strong association with metabolic comorbidities, including hypertension, atherosclerosis, insulin resistance, and prevalent obesity, which may contribute to microvascular alterations in the retinochoroidal system. Yumuşak et al. reported in their study that there was a relationship between BMI and choroidal thickness in obese women. A high BMI is a common condition in PCOS due to central obesity, which occurs as a result of increased insulin resistance, carbohydrate tolerance, and clinical reflection of increased androgen levels. In this case, it may explain the relation of choroidal thickness in PCOS patients in our study, which is consistent with the literature [21]. Similarly, Gonul et al. suggested that CT is positively correlated with BMI and decreases with increasing BMI [22]. Furthermore, our study confirmed a negative correlation between body mass index and choroidal thickness and subfoveal thickness, reinforcing the conclusions of this research. OCTA analysis by Elbeyli et al. showed that patients with PCOS tended to have lower superficial parafoveal vessel density, indicating that the vascular structure of the choroid was affected, supporting our findings [23].

The primary strength of this study is its systematic evaluation of patients with polycystic ovary syndrome (PCOS) categorized into four distinct phenotypes, conducted within a prospective, single-center design. This study has several limitations, such as the small sample size and the inability to assess the long-term effects of PCOS on the retina and choroid due to the relatively young age of the study population. We believe that further studies with larger cohorts including different age groups and long-term follow-up periods will improve our understanding of this issue, which may impact the generalizability of the findings. Further studies employing OCT-A are necessary to identify any choroidal microvascular disorders attributable to PCOS and to assess their relationship with choroidal thickness (CT). Zong et al. investigated the role of genetics in PCOS and showed that women with PCOS have ocular abnormalities, including optic nerve and retinal abnormalities not observed in controls [24]. This can be considered as another limitation of the study. Further studies with genetic analysis and clinical combinations with more participants are important to demonstrate data accuracy.

## 5. Conclusions

In conclusion, PCOS is an endocrine disorder that can impact various organs, including the eye. Its ability to classify patients into distinct phenotypes has facilitated targeted risk analysis. In our study, we demonstrated that PCOS induces changes in the choroid, macula, and retinal nerve fiber layer. While similar studies exist, to our knowledge, comparisons between distinct phenotypes have not been previously conducted. These results may open new avenues for research and garner interest from scholars in the field.

## Figures and Tables

**Table 1 diagnostics-15-00227-t001:** Comparisons of demographics and hormonal data among groups.

Varıable	PCOS Phenotype A *n* = 30	PCOS Phenotype B *n* = 30	PCOS Phenotype C *n* = 30	PCOS Phenotype D *n* = 30	Control Group *n* = 30	*p*-Value
AGE (years ± SD)	28 ± 4.45	27 ± 5.07	27 ± 5.12	26 ± 6.78	29.08 ± 4.96	0.007 ^t^
BMI (kg/m^2^)	26.5 ± 4.9 ^a^	25.7 ± 6.4 ^a^	25.3 ± 5.9 ^a^	25.3 ± 3.9 ^a^	24.8 ± 4.2 ^b^	0.001 ^k^
Estradiol (pg/mL)	87.3 ± 14.94 ^a^	66.5 ± 13.94 ^a^	53.7 ± 16.38 ^a^	45.4 ± 15.61 ^a^	30.6 ± 15.32 ^b^	0.003 ^k^
PG (pg/mL)	1.02 ± 0.73	1.21 ± 0.68	1.43 ± 0.52	1.56 ± 0.48	1.62 ± 0.34	0.087
Free Testosterone (pg/mL)	2.78 ± 2.54 ^a^	1.98 ± 1.78 ^b^	1.72 ± 1.48 ^b^	1.27 ± 0.48 ^b^	0.72 ± 0.36 ^c^	0.002 ^k^

^t^: one-way test of variance, ^k^: Kruskall–Wallis H test, ^a–c^: no difference between phenotypes with the same letter. (Bonferroni-corrected Z test), mean ± s. deviation.

**Table 2 diagnostics-15-00227-t002:** Comparison of the macular and choroidal thickness of the subjects with different phenotypes of PCOS and the control group.

Varıable	PCOS Phenotype A *n* = 30	PCOS Phenotype B *n* = 30	PCOS Phenotype C *n* = 30	PCOS Phenotype D *n* = 30	Control Group *n* = 30	*p*-Value
CSF	266.18 ± 21.88	267.28 ± 22.15	263.34 ± 19.88	266.99 ± 20.24	269.45 ± 22.37	0.06 ^k^
SIM	340.64 ± 15.36	343.48 ± 16.11	345.32 ± 15.98	346.45 ± 16.01	342.61 ± 16.47	0.09 ^k^
TIM	329.96 ± 16.24	332.43 ± 15.87	331.74 ± 17.01	334.66 ± 17.05	333.86 ± 16.76	0.06 ^k^
IIM	343.52 ± 16.11	344.16 ± 16.23	342.91 ± 16.12	344.10 ± 16.15	342.96 ± 15.88	0.08 ^k^
NIM	342.57 ± 17.82	343.07 ± 19.23	344.47 ± 18.26	343.97 ± 19.72	343.47 ± 19.22	0.06 ^k^
SOM	306.24 ± 16.32	305.93 ± 15.98	307.04 ± 16.19	306.78 ± 17.06	307.24 ± 17.01	0.07 ^k^
TOM	286.12 ± 23.34 ^a^	287.86 ± 18.34 ^a^	289.12 ± 17.12 ^a^	287.99 ± 19.15 ^a^	312.56 ± 23.34 ^b^	*p* < 0.001 ^k^
IOM	298.13 ± 16.71	299.03 ± 15.96	298.45 ± 17.01	297.93 ± 15.78	298.7 ± 16.41	0.08 ^k^
NOM	335.62 ± 16.82 ^a^	333.06 ± 17.64 ^a^	334.06 ± 12.32 ^a^	332.96 ± 13.62 ^a^	312.56 ± 15.82 ^b^	*p* < 0.001 ^k^
Foveal center	216.53 ± 14.12	217.15 ± 15.45	215.95 ± 13.43	218.05 ± 14.42	217.56 ± 12.87	0.07 ^k^
CCT	397.5 ± 9.93 ^a^	394.53 ± 9.78 ^a^	393.89 ± 9.68 ^a^	390.84 ± 9.79 ^a^	305.12 ± 5.4 ^b^	*p* < 0.001 ^k^
Nasal CT 500 µ	274.28 ± 5.13 ^a^	272.44 ± 5.23 ^a^	273.37 ± 5.72 ^a^	272.54 ± 5.67 ^a^	256.39 ± 6.23 ^b^	*p* < 0.001 ^k^
Nasal CT 1500 µ	260.46 ± 5.68 ^a^	258.96 ± 5.47 ^a^	259.32 ± 5.62 ^a^	258.92 ± 5.43 ^a^	249.36 ± 5.08 ^b^	*p* < 0.001 ^k^
Temporal CT 500 µ	288.08 ± 6.34 ^a^	286.48 ± 6.53 ^a^	286.52 ± 6.77 ^a^	285.98 ± 6.37 ^a^	276.76 ± 6.34 ^b^	*p* < 0.001 ^k^
Temporal CT 1500 µ	278.53 ± 8.29 ^a^	276.62 ± 8.63 ^a^	276.43 ± 8.04 ^a^	275.43 ± 8.24 ^a^	267.44 ± 8.51 ^b^	*p* < 0.001 ^k^

^k^: Kruskall–Wallis H test, ^a,b^: no difference between phenotypes with the same letter (Bonferroni-corrected Z test), mean ± s. deviation.

**Table 3 diagnostics-15-00227-t003:** The comparison of peripapillary RNFL thickness in women with different phenotypes of PCOS and healthy controls.

Varıable	PCOS Phenotype A *n* = 30	PCOS Phenotype B *n* = 30	PCOS Phenotype C *n* = 30	PCOS Phenotype D *n* = 30	Control Group *n* = 30	*p*-Value
T	84.5 ± 6.43	83.5 ± 6.47	84.56 ± 6.28	83.78 ± 6.42	83.67 ± 6.73	0.07 ^k^
TS	136.25 ± 15.87	135.42 ± 16.27	136.12 ± 16.23	135.72 ± 16.54	135.92 ± 16.51	0.08 ^k^
NS	125.56 ± 13.23 ^a^	125.43 ± 12.41 ^a^	126.13 ± 13.67 ^a^	125.94 ± 12.86 ^a^	112.43 ± 13.53 ^b^	*p* < 0.001 ^k^
N	79.58 ± 13.48 ^a^	79.25 ± 14.24 ^a^	78.95 ± 13.86 ^a^	79.38 ± 14.09 ^a^	74.65 ± 14.28 ^b^	*p* < 0.001 ^k^
NI	131.95 ± 14.94 ^a^	132.86 ± 15.28 ^a^	132.28 ± 15.07 ^a^	132.26 ± 14.96 ^a^	127.85 ± 15.63 ^b^	*p* < 0.001 ^k^
TI	145.25 ± 15.23	144.95 ± 15.78	144.85 ± 14.96	145.15 ± 15.26	144.65 ± 15.84	0.07 ^k^
G	107.05 ± 12.57 ^a^	106.85 ± 15.23 ^a^	106.56 ± 14.34 ^a^	106.95 ± 13.67 ^a^	102.45 ± 15.09 ^b^	*p* < 0.001 ^k^

^k^: Kruskall–Wallis H test, ^a,b^: no difference between phenotypes with the same letter (Bonferroni-corrected Z test), mean ± s. deviation.

**Table 4 diagnostics-15-00227-t004:** The comparison of peripapillary RNFL volumes in women with different phenotypes of PCOS and healthy controls.

Varıable	PCOS Phenotype A *n* = 30	PCOS Phenotype B *n* = 30	PCOS Phenotype C *n* = 30	PCOS Phenotype D *n* = 30	Control Group *n* = 30	*p*-Value
CSF	0.20 (0.19–0.22) ^a^	0.20 (0.18–0.21) ^a^	0.20 (0.19–0.23) ^a^	0.20 (0.18–0.22) ^a^	0.22(0.20–0.24) ^b^	*p* < 0.001 ^k^
SIM	0.53 (0.52–0.56)	0.53 (0.51–0.55)	0.54 (0.52–0.56)	0.54 (0.52–0.55)	0.54 (0.53–0.56)	0.07 ^k^
TIM	0.51 (0.50–0.53) ^a^	0.52 (0.51–0.53) ^a^	0.51 (0.50–0.54) ^a^	0.51 (0.49–0.53) ^a^	0.54 (0.51–0.56) ^b^	*p* < 0.001 ^k^
IIM	0.53 (0.51–0.55)	0.53 (0.51–0.54)	0.54 (0.51–0.55)	0.53 (0.52–0.55)	0.54 (0.52–0.56)	0.01 ^k^
NIM	0.53 (0.52–0.56)	0.54 (0.53–0.56)	0.54 (0.52–0.55)	0.54 (0.50–0.56)	0.53 (0.51–0.55)	0.02 ^k^
SOM	1.62 (1.56–1.68)	1.61 (1.57–1.68)	1.59 (1.56–1.65)	1.60 (1.57–1.65)	1.59 (1.53–1.63)	0.24 ^k^
TOM	1.52 (1.46–1.62) ^a^	1.53 (1.48–1.61) ^a^	1.52 (1.48–1.60) ^a^	1.53 (1.45–1.59) ^a^	1.62 (1.54–1.70) ^b^	*p* < 0.001 ^k^
IOM	1.57 (1.55–1.62)	1.58 (1.54–1.62)	1.56 (1.54–1.62)	1.57 (1.55–1.60)	1.58 (1.54–1.64)	0.19 ^k^
NOM	1.70 (1.64–1.72) ^a^	1.71 (1.63–1.73) ^a^	1.69 (1.66–1.73) ^a^	1.70 (1.66–1.74) ^a^	1.60 (1.54–1.68) ^b^	*p* < 0.001 ^k^
TOTAL VOLUME	8.79 (8.68–8.96)	8.78 (8.66–8.86)	8.78 (8.67–8.95)	8.79 (8.64–8.94)	8.78 (8.66–8.92)	0.23 ^k^

^k^: Kruskall–Wallis H test, ^a,b^: no difference between phenotypes with the same letter (Bonferroni-corrected Z test), mean ± s. deviation.

**Table 5 diagnostics-15-00227-t005:** The Pearson correlation analysis between body mass index, hormonal changes, and choroidal and foveal thicknesses.

	CCT	N500	N1500	T500	T1500	CFT
BMI						
r	−0.175	+0.194	+0.193	+0.187	+0.186	−0.178
*p*	0.013	0.069	0.104	0.423	0.136	0.005
Estradiol (pg/mL)						
r	+0.198	+0.204	+0.198	+0.204	+0.198	+0.204
*p*	0.195	0.136	0.024	0.796	0.856	0.0456
Free Testosterone (pg/mL)						
r	+0.115	+0.217	−0.071	+0.450	+0.326	+0.258
*p*	0.066	0.483	0.166	0.074	0.066	0.073

r: Spearman’s rho correlation coefficient.

## Data Availability

The original contributions presented in this study are included in the article. Further inquiries can be directed to the corresponding author.

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
