# Peer review of "Evaluation of Retinal Changes in Women with Different Phenotypes of Polycystic Ovary Syndrome"

_diagnostics, 2025, doi:10.3390/diagnostics15020227_

Round 1

Reviewer 1 Report

Comments and Suggestions for Authors

The authors need to mention clearly the method of measurement of choroidal thickness.

Whether the measurements were done by more than one observers

Author Response

Thank you very much for considering our manuscript entitled “Evaluation of Retinal Changes in Women with Different Phe-notypes of Polycystic Ovary Syndrome”.

The comments were very constructive and we believe they have helped us improve our manuscript. We have considered all the comments and provided a point by point response in the following pages. Please see the responses below, which are highlighted on the revised version of the manuscript.

1.The authors need to mention clearly the method of measurement of choroidal thickness. Whether the measurements were done by more than one observers.

Thank you very much for your recommendation. Necessary corrections have been made in line with your suggestions, and the choroidal measurement method and whether the measurement was performed by more than one person have been specified.

Reviewer 2 Report

Comments and Suggestions for Authors

I would like to congratulate the authors on their work which is a significant contribution to the literature.

A few suggestions that might improve the quality of the paper:

1. The authors can add how many women were initially recruited and how many were excluded to arrive at the population of 120 women.

2. The authors describe in detail the acquisition method of the retinal scans. They can add that the scans were acquired after dilation with specific type of drops (e.g. tropic amide 1%) and add that the scans were captured during the same time of the day to compensate for any diurnal variation of the choroidal thickness.

3. The authors provide sufficient pathophysiology to explain their findings. They can add some limitation of the study- small sample size limitations related to other confounding factors that might influence retinal thickness etc.

Author Response

Thank you very much for considering our manuscript entitled “Evaluation of Retinal Changes in Women with Different Phe-notypes of Polycystic Ovary Syndrome”.

The comments were very constructive and we believe they have helped us improve our manuscript. We have considered all the comments and provided a point by point response in the following pages. Please see the responses below, which are highlighted on the revised version of the manuscript.

  1. The authors can add how many women were initially recruited and how many were excluded to arrive at the population of 120 women.

The number of people initially included in the study and the number of people with whom the study was completed are indicated.Thank you very much for your valuable contributions.

  1. The authors describe in detail the acquisition method of the retinal scans. They can add that the scans were acquired after dilation with specific type of drops (e.g. tropic amide 1%) and add that the scans were captured during the same time of the day to compensate for any diurnal variation of the choroidal thickness.

Thank you very much for your comments and contributions.Retinal scanning methods and choroidal diurnal measurement time are indicated.

  1. The authors provide sufficient pathophysiology to explain their findings. They can add some limitation of the study- small sample size limitations related to other confounding factors that might influence retinal thickness etc.

The limitations of the study are explained in detail.

Reviewer 3 Report

Comments and Suggestions for Authors

Novel study in dividing polycystic ovary into 4 groups. Methodology including statistical interpretation is very clear.

To further validate their findings and enhance the Discussion we suggest the following minor changes. If the authors cannot add these changes , at least acknowledge N.1 as one of the deficiencies of their paper

1-We need to know the refraction as this will affect the final values in each group

Barsha Lal, David Alonso-Caneiro, Scott A. Read, Andrew Carkeet,

Induced Refractive Error Changes the Optical Coherence Tomography Angiography Transverse Magnification and Vascular Indices,

American Journal of Ophthalmology, Volume 229, 2021, Pages 230-241,

https://doi.org/10.1016/j.ajo.2021.04.012.

2- We need to add references that show vascular changes by OCTA or present a genetic viewpoint

2a-OCTA analysis indicates that patients with PCOS tend to have lower superficial parafoveal vessel densities. Elbeyli, A., Kurtul, B. E., & Karapinar, O. S. (2021). Investigation of the Retinal and Optic Disc Microvascularization in Polycystic Ovary Syndrome: An Optical Coherence Tomography Angiography Study. Ocular Immunology and Inflammation, 31(1), 92–96. https://doi.org/10.1080/09273948.2021.1986546

2b-The parafoveal superior, inferior, and temporal quadrant thickness was significantly higher in the study group compared to control group (p=0.047, p=0.033,and p=0.033, respectively). In patients with PCOS, there were negative correlations between IR and inferior RNFL, total and superior GCC thickness (r=-0.29 p= 0.027, r=-0.27 p=0.050, r=-0.31 p=0.029, respectively).

Yener NP, Ozgen G, Tufekci A, Ozgen L, Aydin GA. Optical Coherence Tomography Angiography Findings in Polycystic Ovary Syndrome. J Coll Physicians Surg Pak. 2021 Sep;31(9):1057-1063. doi: 10.29271/jcpsp.2021.09.1057.

 2c-The results of our retinal segmentation analysis indicate that patients with PCOS tend to have thinner GCL, IPL, and PRL than healthy, age-matched controls due to neurodegeneration likely caused by insulin resistance, or subclinical retinal inflammation.Sirakaya, E., Sirakaya, H. A., Vural, E., Duru, Z., & Aksoy, H. (2020). Determination of Neurodegeneration in Polycystic Ovary Syndrome with Retinal Segmentation Analysis. Current Eye Research, 46(6), 831–838. https://doi.org/10.1080/02713683.2020.1842460

2d-ROLE OF GENETICS: Women with PCOS had eye abnormalities, including abnormalities of the optic nerve, and retina, that were not observed in controls (p = 0.0002). 

Zong, Z., Kalyan, S., Andres, C., Akkor, S., Prior, J. C., & Patel, M. S. (2022). Prevalence of ocular anomalies is increased in women with polycystic ovary syndrome—exploration of association with PAX6 genotype. Ophthalmic Genetics, 43(3), 340–343. https://doi.org/10.1080/13816810.2022.2025605

Author Response

Thank you very much for considering our manuscript entitled “Evaluation of Retinal Changes in Women with Different Phe-notypes of Polycystic Ovary Syndrome”.

The comments were very constructive and we believe they have helped us improve our manuscript. We have considered all the comments and provided a point by point response in the following pages. Please see the responses below, which are highlighted on the revised version of the manuscript.

1-We need to know the refraction as this will affect the final values in each groupBarsha Lal, David Alonso-Caneiro, Scott A. Read, Andrew Carkeet,Induced Refractive Error Changes the Optical Coherence Tomography Angiography Transverse Magnification and Vascular Indices,American Journal of Ophthalmology, Volume 229, 2021, Pages 230 241,https://doi.org/10.1016/j.ajo.2021.04.012.

Thank you very much for your valuable contribution. The refraction values are clearly indicated.

2- We need to add references that show vascular changes by OCTA or present a genetic viewpoint.

2a-OCTA analysis indicates that patients with PCOS tend to have lower superficial parafoveal vessel densities. Elbeyli, A., Kurtul, B. E., & Karapinar, O. S. (2021). Investigation of the Retinal and Optic Disc Microvascularization in Polycystic Ovary Syndrome: An Optical Coherence Tomography Angiography Study. Ocular Immunology and Inflammation, 31(1), 92–96. https://doi.org/10.1080/09273948.2021.1986546

2b-The parafoveal superior, inferior, and temporal quadrant thickness was significantly higher in the study group compared to control group (p=0.047, p=0.033,and p=0.033, respectively). In patients with PCOS, there were negative correlations between IR and inferior RNFL, total and superior GCC thickness (r=-0.29 p= 0.027, r=-0.27 p=0.050, r=-0.31 p=0.029, respectively).Yener NP, Ozgen G, Tufekci A, Ozgen L, Aydin GA. Optical Coherence Tomography Angiography Findings in Polycystic Ovary Syndrome. J Coll Physicians Surg Pak. 2021 Sep;31(9):1057-1063. doi: 10.29271/jcpsp.2021.09.1057.

2c-The results of our retinal segmentation analysis indicate that patients with PCOS tend to have thinner GCL, IPL, and PRL than healthy, age-matched controls due to neurodegeneration likely2caused by insulin resistance, or subclinical retinal inflammation.Sirakaya, E., Sirakaya, H. A., Vural, E., Duru, Z., & Aksoy, H. (2020). Determination of Neurodegeneration in Polycystic Ovary Syndrome with Retinal Segmentation Analysis. Current Eye Research, 46(6), 831–838. https://doi.org/10.1080/02713683.2020.1842460

2d-ROLE OF GENETICS: Women with PCOS had eye abnormalities, including abnormalities of the optic nerve, and retina, that were not observed in controls (p = 0.0002).Zong, Z., Kalyan, S., Andres, C., Akkor, S., Prior, J. C., & Patel, M. S. (2022). Prevalence of ocular anomalies is increased in women with polycystic ovary syndrome—exploration of association with PAX6 genotype. Ophthalmic Genetics, 43(3), 340–343. https://doi.org/10.1080/13816810.2022.2025605

Thank you very much for your valuable contributions. In line with your suggestions, the items you suggested in the discussion section have been added by discussing the results of our study and the literature.